# A top-down approach of sources and non-photosynthetic sinks of carbonyl sulfide from atmospheric measurements over multiple years in the Paris region (France)

**Sauveur Belviso**[1]⊙*, **Benjamin Lebegue**[1]⊙, **Michel Ramonet**[1]⊙, **Victor Kazan**[1]⊙, **Isabelle Pison**[1]⊙, **Antoine Berchet**[1]⊙, **Marc Delmotte**[1]⊙, **Camille Yver-Kwok**[1]⊙, **David Montagne**[2]⊙, **Philippe Ciais**[1]⊙

**1** Laboratoire des Sciences du Climat et de l'Environnement, CEA-CNRS-UVSQ- Université, Paris-Saclay, UMR8212, Gif-sur-Yvette, France, **2** UMR ECOSYS, INRA, AgroParisTech, Université Paris-Saclay, Thiverval-Grignon, France

⊙ These authors contributed equally to this work.
* sauveur.belviso@lsce.ipsl.fr

**Data Availability Statement:** Time series of COS mixing ratio and exchange rates are available at

## Abstract

Carbonyl sulfide (COS) has been proposed as a proxy for carbon dioxide ($CO_2$) taken up by plants at the leaf and ecosystem scales. However, several additional production and removal processes have been identified which could complicate its use at larger scales, among which are soil uptake, dark uptake by plants, and soil and anthropogenic emissions. This study evaluates the significance of these processes at the regional scale through a top-down approach based on atmospheric COS measurements at Gif-sur-Yvette (GIF), a suburban site near Paris (France). Over a period of four and a half years, hourly measurements at 7 m above ground level were performed by gas chromatography and combined with [222]Radon measurements to calculate nocturnal COS fluxes using the Radon-Tracer Method. In addition, the vertical distribution of COS was investigated at a second site, 2 km away from GIF, where a fast gas analyzer deployed on a 100 m tower for several months during winter 2015–2016 recorded mixing ratios at 3 heights (15, 60 and 100 m). COS appears to be homogeneously distributed both horizontally and vertically in the sampling area. The main finding is that the area is a persistent COS sink even during wintertime episodes of strong pollution. Nighttime net uptake rates ranged from -1.5 to -32.8 pmol $m^{-2}$ $s^{-1}$, with an average of -7.3 ± 4.5 pmol $m^{-2}$ $s^{-1}$ (n = 253). However, episodes of biogenic emissions happened each year in June-July (11.9 ± 6.2 pmol $m^{-2}$ $s^{-1}$, n = 24). Preliminary analyses of simulated footprints of source areas influencing the recorded COS data suggest that long-range transport of COS from anthropogenic sources located in Benelux, Eastern France and Germany occasionally impacts the Paris area during wintertime. These production and removal processes may limit the use of COS to assess regional-scale $CO_2$ uptake in Europe by plants through inverse modeling.

https://mycore.core-cloud.net/index.php/s/
XwULEXsgwFM1Tus/download.

**Funding:** The authors received no specific funding
for this work.

**Competing interests:** The authors have declared
that no competing interest exist.

## Introduction

Measurements of carbonyl sulfide (COS) in the troposphere could help better assess how much carbon dioxide ($CO_2$) is taken up by land photosynthesis. For example, Launois et al. used atmospheric COS data to constrain the annual, seasonal and spatial variations of the gross primary production (GPP) of three dynamic global vegetation models [1]. Their approach allowed for bias recognition in the GPP representation of the three selected models, both in terms of annual global total carbon fluxes, and also in the phase and amplitude of the seasonal variations. In their concise review of the reasons why COS could be used as a tracer for photosynthetic uptake, which are based on aspects of plant physiology [2–3], leaf scale measurements (e.g., [4]), measurements of ecosystem fluxes (e.g., [5]), and joint surveys of atmospheric concentrations of COS and $CO_2$ mainly in the US (e.g.[6]), Campbell et al. also pointed to additional processes that could complicate the use of COS as a quantitative tracer of large scale (regional or continental) GPP [7]. These processes are soil uptake, dark uptake by plants, and soil and anthropogenic emissions. Night COS plant uptake is always a complicating factor that has to be accounted for in COS applications at the regional and global scales. At the ecosystem scale, the night and day fluxes can be measured directly. However, at the regional scale, separating night and day fluxes is not possible for records of background COS concentration in the atmosphere, where the signal of day and night sources and sinks is mixed by transport before reaching the point of observation. For a more extensive review of ecosystem processes and anthropogenic sources of COS, the reader is referred to the recently released articles by Whelan et al. who stressed the need for more year-round measurements from a larger number of biomes to provide reliable estimates of terrestrial COS fluxes [8], and Zumkehr et al. who provided a global gridded anthropogenic emissions inventory of COS [9].

Following the approach described by Belviso et al. [10] now applied on a much longer time-period of about four and a half years, continuous atmospheric COS and $^{222}$Radon measurements were collected in a suburban area (Gif-sur-Yvette) in the Greater Paris Area, at two measurement sites. These data are used to quantify non-photosynthetic uptake and ecosystem emission of COS. Nighttime fluxes have been computed from the now widely used Radon-Tracer Method (RTM) as by Yver et al. [11], Lopez et al. [12], Belviso et al. [10] and Kooijmans et al. [13] for molecular hydrogen, nitrous oxide and COS, respectively. The Greater Paris Area occasionally faces intense pollution episodes during stable atmospheric conditions, notably during winter (e.g. [14]). COS and $^{222}$Radon measurements were also used to assess the significance of Greater Paris Area anthropogenic emissions.

In the first section, we describe the sampling sites and the analytical methods used for estimating fluxes using mixing ratio observations. Multi-year, multi-height and multi-site observations are described in the second section. We also discuss the potential implications of these results for using COS to assess $CO_2$ uptake by plants in European countries.

## Materials and methods

No specific permissions were required for these locations/activities. I confirm that the field studies did not involve endangered or protected species.

### Site description

The two measurement sites are located 2 km apart at Gif-sur-Yvette (GIF) and Saclay (SAC), about 20 km to the south west of Paris, as shown in Fig 1.

The land cover classification was retrieved from an online application at https://www.theia-land.fr/en/product/land-cover-map/. The Geotiff file is named OCS_2016_CESBIO.tif. For more information, see http://osr-cesbio.ups-tlse.fr (in French). At the local scale, the two sites

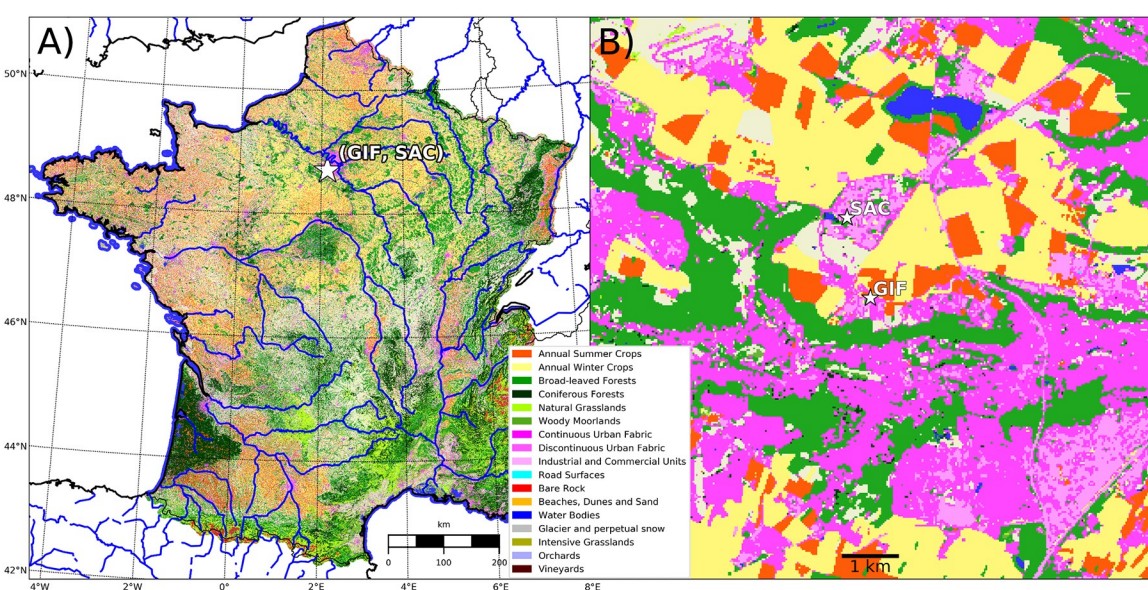

**Fig 1. Land cover map showing the location of the sampling sites (GIF, SAC).**

are surrounded by roads and a highway, urban areas, croplands, grasslands, lawns (including a golf course between SAC and GIF) and forests to the west and south. Building and tree height does not exceed 7 m at the GIF site. Taller buildings exist at the SAC site but not in the immediate vicinity of the sampling site.

The cultivated soils are classified as well-drained to stagnic Luvisols [15]. They developed on de-carbonated quaternary loess deposits and lay over a millstone-clay layer impeding natural drainage. These soils are thus frequently waterlogged during winter, i.e. from December to March. They showed the classical ABtC horizonation with surface A horizon composed of 19% clay, 75% silt and 6% sand and subsurface Bt horizon composed of 30% clay, 66% silt and 4% sand. The ploughed horizon of these Luvisols showed neutral pH (7.6) and relatively low organic carbon and nitrogen content (respectively 10 and 1 g kg$^{-1}$). By contrasting, the forested soils are not waterlogged except locally and their topsoil horizons are sandier with contents up to 70 to 90%, richer in organic carbon and nitrogen with contents up to 50 and 3 g kg$^{-1}$ respectively and much more acidic (pH lesser than 5).

The Paris region experienced several air pollution episodes during the four and a half year atmospheric COS record. The history of pollution episodes is recorded by the AIRPARIF air quality association and is available in French at https://www.airparif.asso.fr/alertes/historique. The four major compounds measured for the Paris pollution index are nitrogen dioxide ($NO_2$), sulfur dioxide ($SO_2$), ozone ($O_3$), and fine particles less than 10 μm in diameter ($PM_{10}$). Two thresholds are defined for the air quality index, "information (I)" and "alert (A)". All periods classified as "I" or "A" were counted as pollution for the COS time series. The definitions of thresholds can be found in a table named "Tableau des seuils" by clicking on the above URL. For $PM_{10}$, the threshold for "A" is an average daily concentration in excess of 80 μg m$^{-3}$, while for "I" the threshold is 50 μg m$^{-3}$.

## COS and $^{222}$Radon measurements

**COS measurements by GC at the GIF site.** At the GIF site, all sampling lines were made of Synflex tubing which is suitable for COS analysis [13]. Air was sampled 7 m above ground

level on the top of a building for about four and a half years, and analyzed on an hourly basis by gas chromatography (GC). The COS analytical measurement technique is described in [16]. In short, COS is trapped and desorbed using an automated sampling system (Entech P7100 preconcentrator), then analyzed by gas chromatography with pulsed flame photometry detection (PFPD, Varian Model 3800 [16]). Central to this study is the long-term repeatability (LTR) of COS expressed through the standard deviation (SD) of 96 averaged measures over 4.5 years of atmospheric air from a compressed cylinder prepared and calibrated by NOAA-ESRL (long-term air standard), containing 448.6 ± 0.2 ppt of COS (S1 Fig). LTR assessed to be of 8.6 ppt or 1.9% in relative.

Calibration was performed about every three weeks using a calibration gas supplied by Air Products [16] and the drift between calibrations was assessed on a weekly basis using a short-term target gas. Our own calibration scale is highly consistent with the NOAA-2004 scale as shown in S1 Fig where the difference between the assigned value of the NOAA long-term air standard and the average of 96 analyses of that standard is smaller than 1%. It is worth noting that not all natural air standards prepared in Aculife-treated aluminum cylinders and calibrated by NOAA-ESRL exhibit such a high stability over a period of several years. Drift rates can be as high as 5% yr$^{-1}$. We did not identify the mechanism responsible for COS production in some cylinders and not in the others. Today, NOAA-ESRL recommends stainless steel cylinders for COS (B. Hall pers. com. 02/14/2019).

**Computation of COS fluxes with RTM at the GIF site.** $^{222}$Radon, a tracer uniformly emitted by soils whose atmospheric variations near the surface are mainly caused by variations of planetary boundary layer height (PBLh) [17], was monitored at a height of 7 m at GIF. During nocturnal temperature inversions, the accumulation of $^{222}$Rn in the boundary layer results from a lowering of vertical mixing [10]. Hence, in the absence of PBLh estimates from lidar measurements, the PBLh effect on observations was assessed using $^{222}$Rn measurements. $^{222}$Radon was measured using the active deposit method, that is, via the radioactive decay of its daughters attached to aerosols, and was calibrated against an Australian Nuclear Science and Technology Organisation (ANSTO)-built detector for continuous monitoring of radon concentration in air [18]. Flux computations using the RTM are described in detail in [10–11]. In short, the RTM estimates nocturnal fluxes using (1) hourly COS mixing ratio observations, (2) time-averaged $^{222}$Rn values reported at the beginning of the two hour averaging interval, and (3) estimates of $^{222}$Rn local exhalation rates as by [11]. Fluxes are computed essentially as the product of COS/$^{222}$Rn slopes and $^{222}$Rn local exhalation rates. Data that are used to calculate the linear regression slope during night-time inversion (COS/$^{222}$Rn) are selected according to the time of day as follows: [18; 6 [(UTC) in spring, [19; 5 [in summer, [17; 6 [in autumn and winter. In order to be considered significant, the nocturnal COS concentration increases or decreases and $^{222}$Rn concentration increases have to be greater than 15 ppt and 0.5 Bq m$^{-3}$, respectively, and the coefficient of determination of the linear regression (r$^2$) should be greater than 0.6. Two examples of fluxes, shown with mixing ratio and concentration measurements and with r$^2$ values, are displayed in S2 Fig for the nights of April 11, 2015 (negative flux, sink of COS) and July 9, 2016 (positive flux, source of COS).

**COS measurements by QCL at the SAC site.** At the SAC tower, all sampling lines were made of Synflex tubing. COS was measured at 3 heights (15 m, 60 m and 100 m) with a fast gas analyzer (Quantum Cascade Laser, mini QCL COS/CO$_2$/H$_2$O, Aerodyne Res.) for 6 months in 2015–2016 covering winter and spring pollution episodes. Before being deployed at the SAC site, the specifications of our mini QCL instrument were evaluated in the laboratory following the standardized testing protocols designed for analyzers measuring H$_2$O, CO$_2$, CH$_4$, CO and N$_2$O in the framework of the ICOS (Integrated Carbon Observation System) European Infrastructure [19–20]. We determined (1) the continuous measurement repeatability (CMR) for

COS, (2) the optimal averaging time estimated by using Allan standard deviation plots, (3) the stabilization time, (4) the short-term repeatability (STR), (5) the long-term repeatability (LTR, in two different ways), (6) the temperature dependence, (7) the water vapor sensitivity, (8) the linearity, and defined a calibration strategy. The reader is referred to Lebegue et al. [20] for a detailed description of each test and to the series of figures provided in the supplementary material, the legends of which contain other technical details (S3–S11 Figs). The QCL's measurement frequency is 1 Hz. When the instrument was used without "zero" air spectrum measurements (i.e. without introducing highest purity nitrogen into the sampling cell), which is the only way to get true CMR measurements, the instrument drifted at a huge rate of about 180 ppt d$^{-1}$ and the Allan deviation increased from 2.4 ppt at 10 s to 24.1 ppt at 10$^4$ s averaging time (S3 Fig). In this configuration, the instrument could not cope with fluctuations in laser light intensity. When "zero" air spectrum was measured once every 20 minutes, as recommended by the manufacturer, the drifting rate was considerably reduced (0.8 ppt d$^{-1}$) and the Allan deviation was systematically better than 5.6 ppt and decreased down to 1.1 ppt at 40 s averaging time (S3 Fig). In that case, the instrument coped well with the fluctuations in laser light intensity. The time necessary for the instrument to reach a stable value when changing the sample analyzed was three minutes at the most (S4 Fig). The STR for COS was 3.6 ppt and the amplitude of variations peak-to-peak was 13.6 ppt (S5 Fig). The LTR was 5.1 ppt and the amplitude of variations peak-to-peak was 20.7 ppt (S6 Fig). The sensitivity of the mini-QCL to applied large room temperature variations (20.7–32.5˚C, by increasing heating in the room) was rather high (about 80 ppt peak-to-peak) but no linear dependence was found between COS variations and cell temperature changes (S7 Fig). In other words, it is not possible for the user to add an instrumental specific correction that could be applied to the final data when large cell temperature changes occur. The instrument required an extra humidity correction in addition to the one already applied by the manufacturer, as shown in S8 Fig where the difference between humid and dry COS mole fractions is plotted against the sample's water vapor percentage. A humidifying bench was used to give precise control of the sample's water vapor percentage. The empirical vapor correction is applied systematically to raw data as follows:

$$[\text{COS}]_{\text{dry}} = [\text{COS}]_{\text{humid}} - (1 - 2.7[\text{H}_2\text{O}] - 4.1[\text{H}_2\text{O}]^2) \tag{1}$$

with [COS] and [$\text{H}_2\text{O}$] in ppt and % volume, respectively.

The linearity of the mini-QCL was assessed using either ambient air measurement with the mini-QCL next to the GC (S9 Fig), or using calibration tanks with known COS mole fractions (three air compressed cylinders analyzed by GC including the NOAA-ESRL calibration tank used for the LTR assessment of GC measurements). In May 2015, the instruments captured the same episodes of low and high COS levels in the range 400–610 ppt (GC analysis, S9A Fig). GC and laser data, after water vapor correction, were highly correlated ($R^2$ = 0.92, n = 252, P ≪ 0.05, 95% confidence interval of the slope = 0.88–0.95, S9B Fig). Using three calibration tanks with COS content in the range 452.3–729.6 ppt (from the GC analysis), the slope of the QCL vs GC regression line was 0.95 ($R^2$ = 0.999, n = 3, P = 0.015) with a 95% confidence interval in the range 0.65–1.24, and the ordinate (- 1 ppt) was not significantly different from zero (S10 Fig). Hence, the response of the mini-QCL analyzer was linear over the observed range 390–690 ppt, which almost spans the full tropospheric range, but raw data had to be calibrated by applying the multiplying factor 1.0547 obtained with the curve forced through zero. Certified calibration gases spanning the full tropospheric range and showing long-term stability are difficult to obtain. That is why it was impossible for us to use at least three calibration gases, as recommended within the ICOS program [19], to check the variations in the long-term of the response curve of the mini-QCL for COS. Instead, the following method of calibration was

used:

$$[COS]_{final} = 1.0547 \text{ x } [COS]_{dry} + B \tag{2}$$

with the value of the bias B obtained from the analysis with the mini-QCL of a calibration tank (CAL, dry gas analyzed about every fifteen hours) with COS assigned from GC-MS and GC-PFPD measurements (certified data) at the beginning of the survey:

$$B = [COS]_{certified} - 1.0547 \text{ x } [COS]_{CAL} \tag{3}$$

The drift of the instrument was then assessed on at least a daily basis using a target gas, as was done to determine the LTR but on a longer period as shown in S11 Fig when the instrument was operated at the SAC site. The LTR for target gas measurements measured approximately daily over about 8 months was 6.8 ppt with an interquartile range of 6.7 ppt and a min-max range of target gas values of 50 ppt.

In field conditions, the averaging time of a twenty minute long injection was 15 minutes. In other words, a COS mean value ± 1SD is calculated for each injection of atmospheric air by taking the last 15 min of each analysis.

### Backward trajectories and footprint computations

Three-day backward trajectories were calculated and downloaded from the HYSPLIT web site (https://ready.arl.noaa.gov/hypub-bin/trajtype.pl?runtype=archive, [21]) at the location of the GIF site (100 m agl and 260 m asl) using HYSPLIT's normal mode. A new trajectory was calculated every 24 hrs at noon and a maximum number of 7 trajectories was shown for better readability. Three-day backward trajectories were also computed using the FLEXPART Lagrangian particle dispersion model version 9.2 [22], driven by meteorological fields from the European Centre for Medium-Range Weather Forecasts (ECMWF) operational products at 1˚ x 1˚ spatial resolution [23]. Footprints were computed with FLEXPART by releasing a batch of 1000 neutral inert air tracer particles for each date at 12:00 UTC. An ensemble of particle batches was also released to mimic HYSPLIT's "ensemble" mode [21]. Two dimensional footprint maps were computed for each grid point by summing the number of particles being transported within the boundary layer above the given location. FLEXPART's daily footprints in ensemble mode will be shown in the main body of the text whereas HYSPLIT's backward trajectories in normal mode over several days are supplied as supplementary figures for completeness.

## Results and discussion

### Multi-year variations of COS mixing ratio and exchange rates

The first dataset (August 2014 to July 2019) is made of over 30,500 hourly measurements of atmospheric COS mixing ratio at the GIF site, 7 m above ground level (Fig 2A).

Monthly data generally rely on several hundreds of hourly measurements (n = 527 on average over 56 months) with a few exceptions such as August 2014 (n = 76) and October 2017 (n = 185) due to gaps in the time series in these months (Fig 2B). The largest peak of COS was recorded in June 2016 with values above 600 ppt (Fig 2B). COS presents clear seasonal variations with a maximum generally in spring and a minimum in autumn (Fig 2B). Since the present study aims at identifying the sources and sinks of COS in the footprint area, the question of how these processes impact the shape of the mean seasonal cycle, its amplitude and the sea-land gradient will be addressed subsequently (manuscript in preparation).

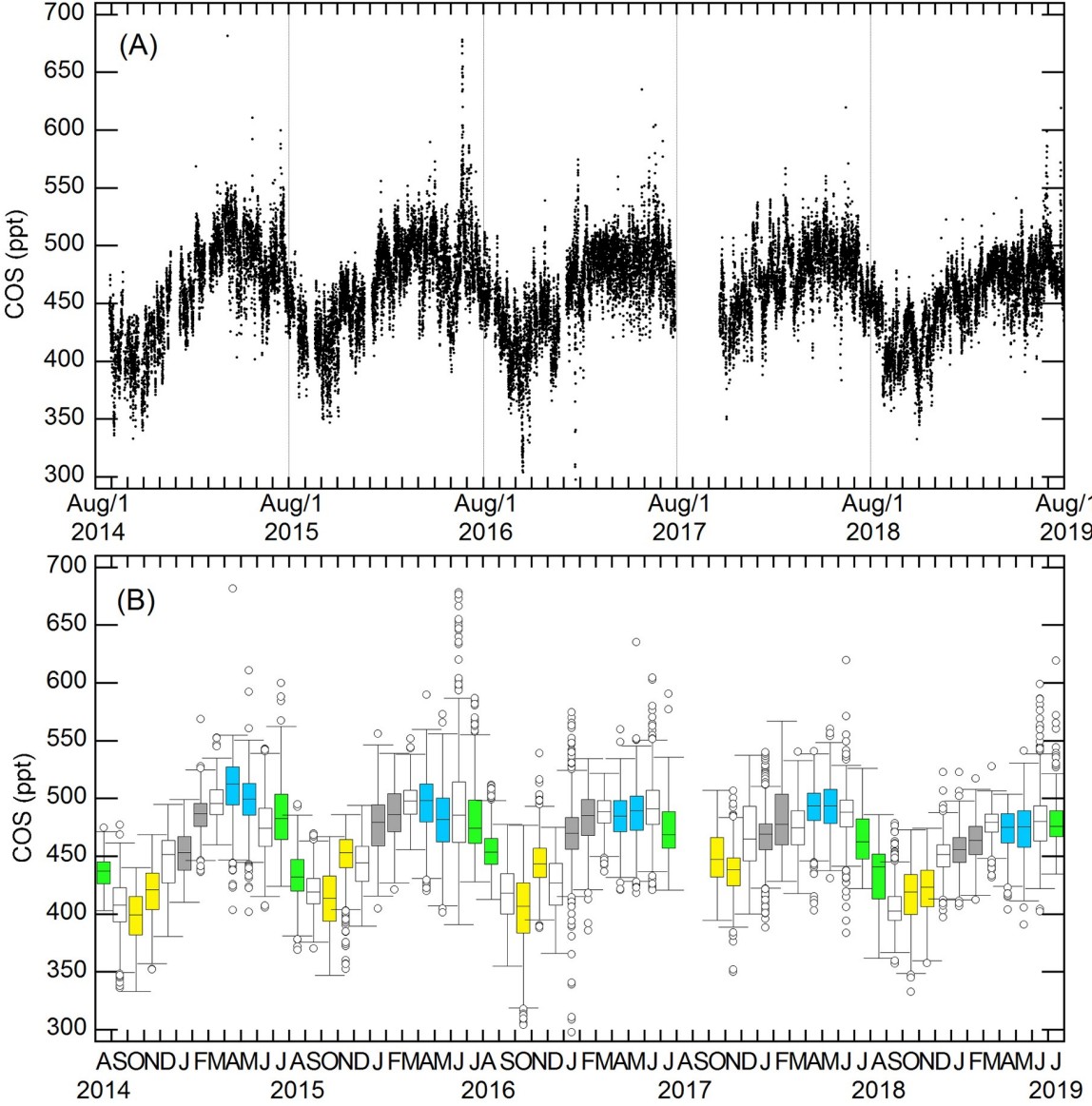

**Fig 2. Multi-year variations in COS monthly mean mixing ratio at GIF.** Panel (A) shows the full record of hourly measurements, with the data gap in summer and early autumn 2017 being a failure of the Entech preconcentrator. Basic statistics (median, upper and lower quartiles, outliers) of monthly data are shown in panel (B). Monthly data for winter (January-February), spring (March-April), summer (July-August) and autumn (October-November) are displayed in grey, blue, green and yellow, respectively. Transition periods are shown in white.

The Radon-Tracer Method allowed us to make 277 determinations of nocturnal fluxes of COS throughout the 2014–2018 period (Fig 3).

90% (n = 253) of nocturnal Radon accumulation episodes coincided with a decrease of COS, indicating a net sink in the footprint of the site. Given the sampling frequency (hour) and the winds in the 277 nocturnal episodes (interquartile of wind speed at 10 m height from 2.6 to 6.2 km h$^{-1}$, S12 Fig) we estimate this footprint to be typically within 4.5 km of the station. Nighttime net uptake rates from the radon method ranged from -1.5 to -32.8 pmol m$^{-2}$ s$^{-1}$, with a median value of -6.4 pmol m$^{-2}$ s$^{-1}$ (interquartile from -4.0 to -8.9 pmol m$^{-2}$ s$^{-1}$; see S13A Fig). The median COS sink was -6.4, -5.9 and -6.9 pmol m$^{-2}$ s$^{-1}$ during winter (December–

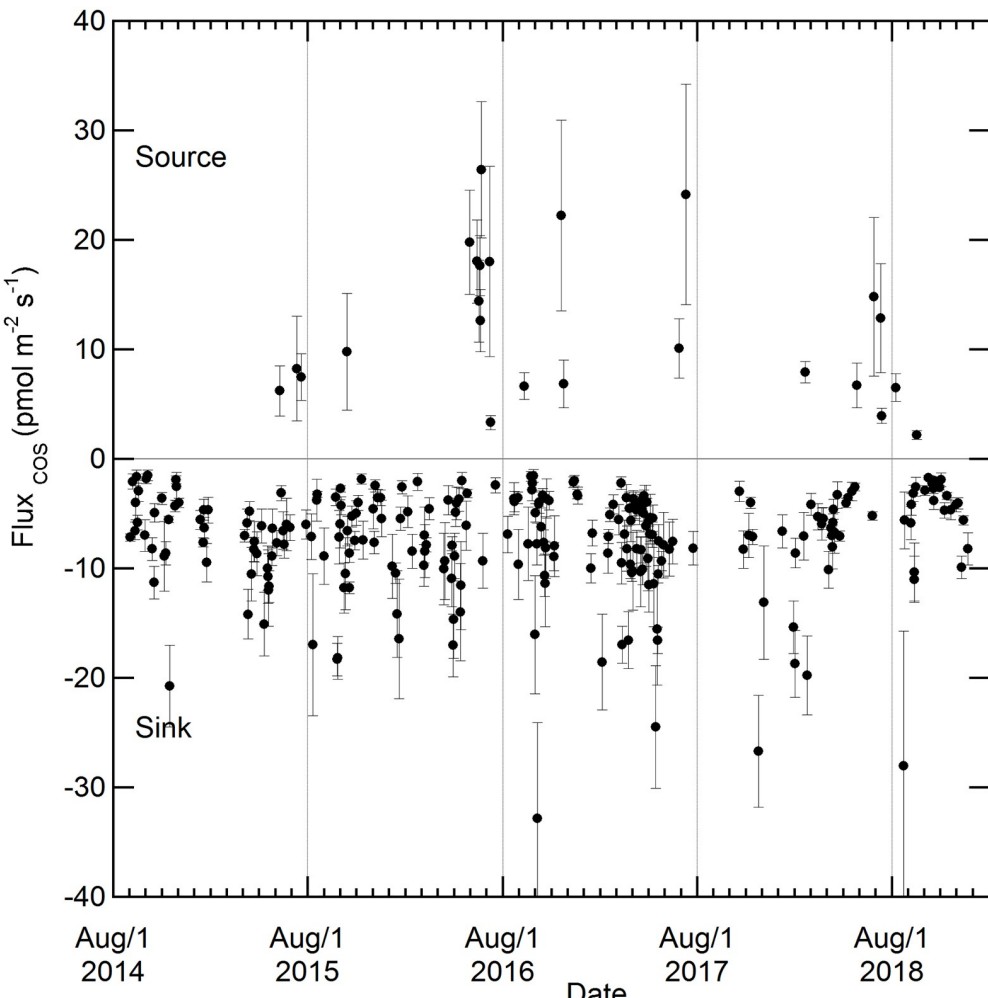

**Fig 3. Multi-year variations in COS exchange rates at the GIF site.** These are nocturnal fluxes obtained by Radon Tracer Method. The uncertainties in the COS flux come from the regression COS/Rn.

March), autumn (September–November) and spring-summer (April–August), respectively (S13B Fig). Since the median value of the COS non-photosynthetic sink is only 8% higher in spring-summer than during the winter (S13B Fig, not significant at $p < 0.05$, two-tailed Mann-Whitney U Test), a higher nocturnal uptake by plants in the growing season may not be the main contribution to the observed COS drawdown at night, thus unlike observed over some forest ecosystems [24, 25]. Rather, soil uptake should be the dominant sink process [8] during nighttime [26–28].

The average emission rate associated with the 24 nocturnal emission events recorded during about four years was of 12.0 pmol m$^{-2}$ s$^{-1}$ (median = 10.0 pmol m$^{-2}$ s$^{-1}$, IQR = 6.7–17.5 pmol m$^{-2}$ s$^{-1}$ (S13A Fig)) with 18 of these events being in summer months and 5 in autumn (Fig 3). In general, concomitant Radon and COS enhancements are observed during periods where nocturnal winds are not oriented from Paris city where one would expect COS to be emitted from anthropogenic sources (S14 Fig). Hence, data suggest that the area near GIF influencing nocturnal changes seasonally switched from sink to source, mostly in the June and July summer months. This seasonality suggest that these COS emission events are biophysically controlled rather than anthropically-controlled, as the latter are prone to be either

relatively constant through time or, by contrast, highly episodic. The addition of fresh crop residues [29, 30], high incoming solar radiation, high soil surface temperature, dry soil conditions [31–33, 8] or, by contrast, waterlogging [34, 8] are among the environmental conditions frequently reported to enhance COS emissions from soils. However, the addition of crop residues could not explain COS emissions recorded in June and early July before the harvest of main crops. Similarly, high incoming solar radiation, high soil surface temperature or dry conditions (or waterlogging), could not explain autumn (or summer) COS emissions. COS emissions seem thus driven by a combination of several processes that still need to be identified and quantified in the GIF area. Additionally, the role of agricultural fields should be investigated because some crops like oilseed rape, which are grown in the GIF area, have recently been shown to release COS in certain conditions as in response to fungal infection [35].

## Anthropogenic emissions of COS from and outside the Paris region

The Paris region is frequently subject to large-scale particulate matter (PM) pollution episodes, notably during winter and spring (e.g. [14]). To give an illustration of the COS atmospheric signature during such pollution episodes, we extracted from the GC and mini-QCL time series four sets of observations to zoom into periods of pollution events lasting several days. A first severe PM pollution episode occurred during February 2015 (Fig 4A).

The PM concentration exceeded the "Alert" threshold defined by AIRPARIF on February 12 because stable nocturnal and diurnal atmospheric conditions prevailed that day, as shown by radon data (Fig 4A). The enhancements in PM and $^{222}$Rn were associated with COS losses. Surprisingly, during another intense PM pollution episode that took place in March 2015 and was extensively described in [14], both time series, especially in March 17 and 18, were more in phase (Fig 4B). A similar approach was applied in winter 2016 with both COS instruments running in parallel at a distance of about two kilometers (Fig 5).

Although there were data gaps in the Dec-Jan time series, very few gaps remained when the GIF and SAC data were combined. In general, there were small spatial variations and stratification in COS across the two sites and the 93 m vertical gradient, at most a few tens of ppt (Fig 5A). Only one of the many $^{222}$Rn accumulation events in December 2015 and January 2016 coincided with a build-up of COS near the surface. This single event took place on 20 and 21 January 2016 during a medium intensity PM pollution event, i.e. over the "Inform" threshold. The March 2016 survey was also carried out with both COS instruments running in parallel at a small separation (Fig 5B). One of the three medium intensity PM pollution events recorded during that period coincided with a clear rise in COS from an average background of 485 ppt (GC data, March 10) up to about 540 ppt (March 11). The characterization of several tens of PM pollution events defined by AIRPARIF over 4.5 years and the application of a multi-tracer approach to the sources and sinks of COS as by [10, 36] will be addressed in a manuscript in preparation. Nevertheless, from the four illustrations provided in Figs 4 and 5 it is possible to draw preliminary conclusions as to the importance of anthropogenic emissions of COS in the Greater Paris Area. In general, a drawdown of COS is observed during stable atmospheric conditions favorable to the occurrence of PM pollution episodes in winter and spring. This is a strong indication that the anthropogenic emissions of COS are offset by natural uptake because the juxtaposition of anthropogenic sources and natural sinks at ground level in the Greater Paris Area allow fluxes to be transported and mixed laterally within the atmospheric boundary layer during inversion episodes of several day duration. Hence, the recent inventory of primary and secondary sources of anthropogenic COS provided by Zumkehr et al. [9] deserves major revision in our area. Indeed, a natural uptake of roughly -105 pmol m$^{-2}$ s$^{-1}$ would be required to offset the total anthropogenic COS emissions put forward by [9], i.e. 98

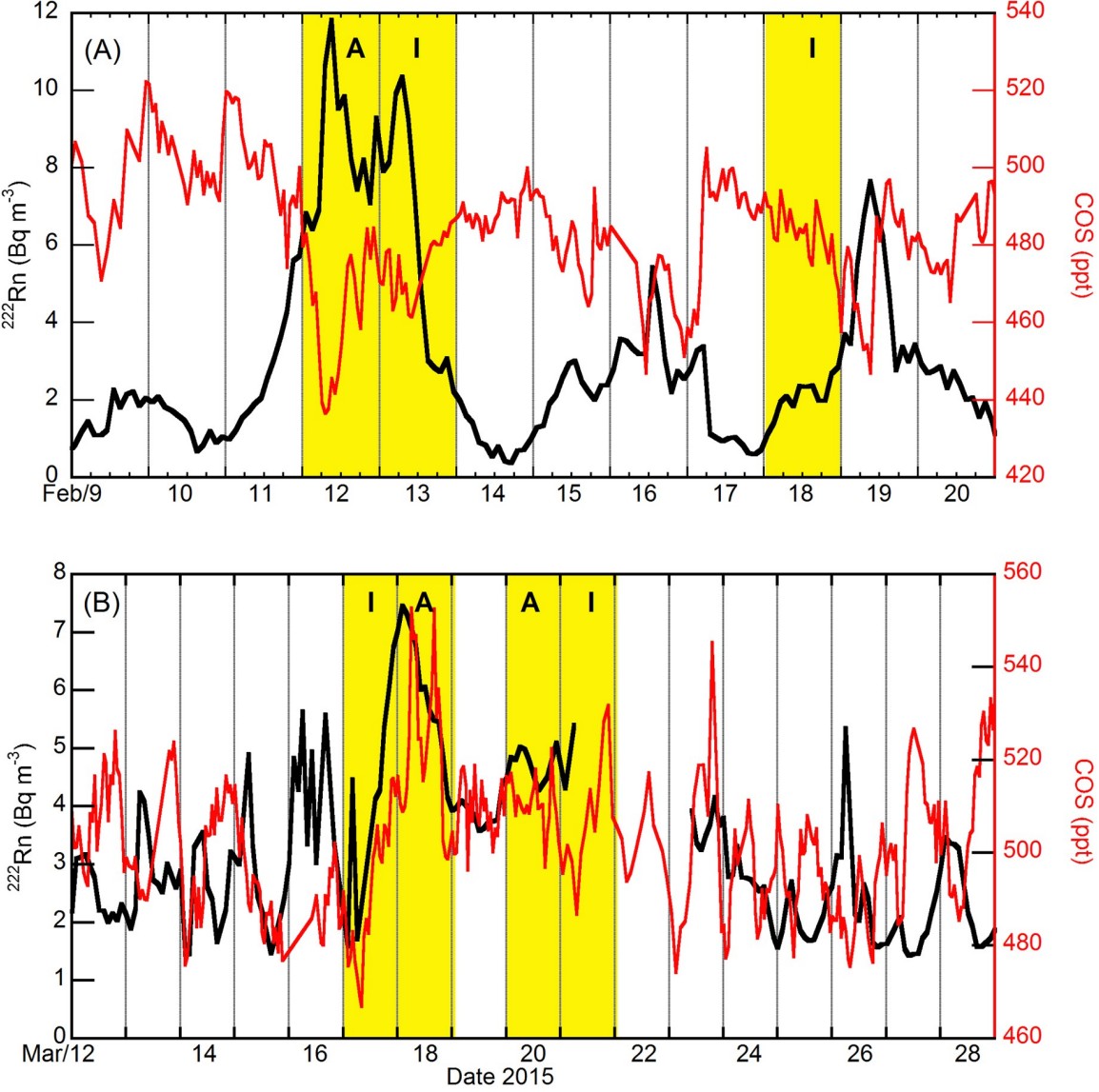

**Fig 4. Hourly variations in $^{222}$Rn activity and COS mixing ratio at the GIF site.** (A) For February and (B) March 2015. The red and black lines correspond to COS and $^{222}$Rn data at GIF (7 m), respectively. The yellow bands correspond to PM pollution episodes over the "**I**nforming" or the "**A**lert" thresholds defined by AIRPARIF.

pmol m$^{-2}$ s$^{-1}$, to get a net flux in the range from -4.0 to -8.9 pmol m$^{-2}$ s$^{-1}$ (S13A Fig). The origin of the fewer events of COS build-up during PM pollution episodes begs question. A preliminary analysis relying on selected back trajectories of air masses suggest that anthropogenic COS is advected over the Paris region when air masses are traced from Eastern France, Benelux and Germany as shown in Fig 6B–6D.

Indeed, when FLEXPART 3-day footprints in ensemble mode point towards the SW and the center of France, as well as towards the Paris region itself (Fig 6A), the PM pollution peak observed on February 12, 2015, is depleted in COS (Fig 4A). Note that contrary to FLEXPART, HYSPLIT in normal mode ending February 12 shows a contribution from the Central-East sector of France (S15A Fig). Data show that when FLEXPART 3-day footprints in ensemble

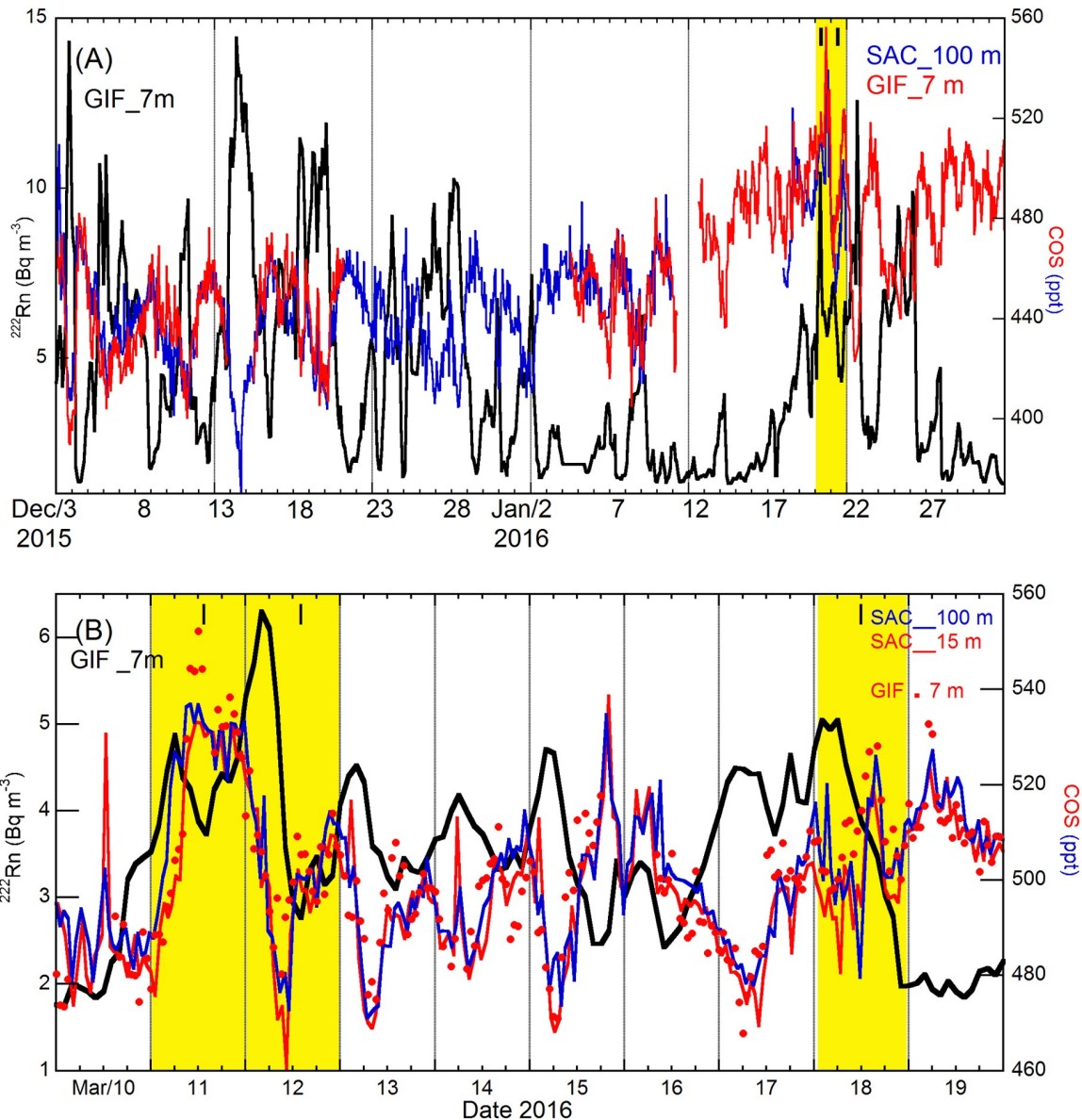

**Fig 5. Hourly variations in $^{222}$Rn activity and COS mixing ratio at the GIF and SAC sites.** (A) For December 2015 –January 2016 and (B) March 2016. The blue and black lines correspond to COS data at SAC (100 m) and $^{222}$Rn data at GIF (7 m), respectively. SAC data at 15 m are not shown because there was no significant vertical gradient between 15 m and 100 m. The red line in panel A and the red dots in panel B correspond to COS data at GIF. The red line in panel B corresponds to COS data at SAC (15 m). The yellow bands correspond to PM pollution episodes over the "**I**nforming" or the "**A**lert" thresholds defined by AIRPARIF.

mode point towards Eastern France (Fig 6C), Benelux and Germany (Fig 6B and 6D), PM pollution peaks and COS enhancements occur concomitantly (Figs 4B and 5). However, when remote transport of COS anthropogenic emissions take place, the COS signal can lag $^{222}$Rn signal rather than being in phase (Figs 4B and 5B) because COS can be removed during transport by biogenic sinks at ground level and $^{222}$Rn accumulate concomitantly. The existence of a phase difference between COS and $^{222}$Rn signals prevents these events from being detected and quantified as local emission events using the RTM.

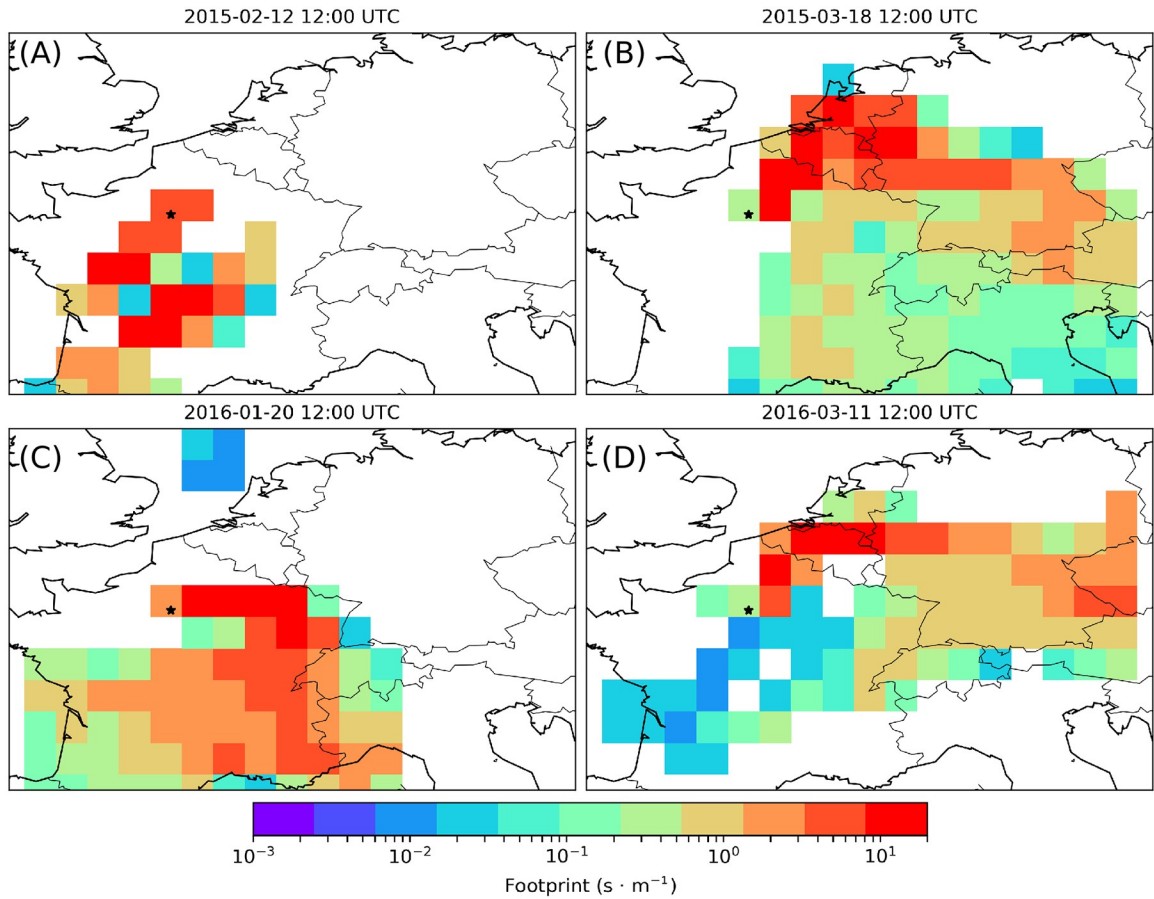

**Fig 6. Selected FLEXPART's footprints in ensemble mode during four pollution events.** These are for the pollution events dated February 12, 2015 (A), March 18, 2015 (B), January 20, 2016 (C) and March 11, 2016 (D). Selected back trajectories ending at GIF at 12:00 UTC, computed 100 m agl using HYSPLIT's normal mode, are shown for comparison in S15–S17 Figs.

In their multi-site approach at characterizing the intense PM pollution episode in March 2015, Petit et al. [14] calculated clusters of HYSPLIT trajectories at the SIRTA sampling site located about 4 km East of the GIF site. The mean trajectory associated with the first pollution episode, dated March 17–18 (see Fig 2 in [14]), shows that an air mass was advected over the Paris region from the NE (i.e. Germany and Luxemburg). The second episode, dated March 20–21, was associated with a different cluster and a mean trajectory oriented N (i.e., North Sea and coastal Belgium, see also S15B Fig). Hence, continental air masses advected over the Paris region from the NE were about 40 ppt richer in COS than air masses advected from the N.

Backward trajectories at 100 m above ground level calculated using the HYSPLIT's normal mode, show that the transport of the air masses sampled during January 20, 2016, when COS levels were in the range 500–550 ppt occurred at low altitudes (below about 200 m agl) with an origin from the East, more specifically from Southern Germany (S16 Fig). FLEXPART 3-day footprints in ensemble mode point towards the East center of France up to the Swiss border (Fig 6C). By January 23, the air mass sampled at GIF / SAC was advected from the Southwest and passed over Northern Spain and the Gulf of Biscay (S16 Fig). It displayed low levels of $^{222}$Rn (Fig 5A), a typical signature of marine air that has not been in close contact with the ground during transport over Western France (S16 Fig). The COS content of this marine air mass being 500 ± 10 ppt, we can conclude that the COS enhancement of polluted continental

air relative to marine air did not exceed a few tens of ppt during that event (Fig 5A). During December 2015 and early January 2016, however, $^{222}$Rn increases of the same magnitude as during the late January 2016 pollution episode were associated with COS losses, their amplitudes being in the range 30–100 ppt (Fig 5A). A return to rather clean continental air conditions was observed in January 24 and 25. Hence, the COS signature of polluted continental air relative to clean continental air in this period was higher than 50 ppt and reached up to 100 ppt.

The last pollution episode associated with COS enhancements, observed in March 2016 (Fig 5B), very much resembled that in March 2015. Backward trajectories (S17 Fig) and footprints (Fig 6D) indicate that when polluted air is advected from western Germany and Benelux its COS content is 50–60 ppt higher than that of marine air.

This raises the question of the deployment of atmospheric COS monitoring stations in Europe for carbon cycle application. A move towards the East would be required to reduce the marine influence and increase that of the vegetation but sites located more inland (e.g. in Germany and Central Europe) would likely be more impacted by biogenic and anthropogenic emissions especially during summer. It is worth to remind that anthropogenic influences were also reported in June 2013 at the Observatoire de Haute Provence a rural site in Southern France [16]. To guide the decision of where to install new COS monitoring stations in Europe for carbon cycle application, we stress the need of (1) updating the inventory of primary and secondary anthropogenic COS sources, (2) improving our understanding of summer biogenic emissions and (3) assessing the footprint of stations prior to the deployment of COS instruments.

## Conclusions

A four and a half year time series of near surface atmospheric COS concentrations and fluxes calculated for nocturnal situations of low boundary layer height using Radon-222 measurements is provided. There is compelling evidence that the anthropogenic fluxes from the Paris area make a much smaller contribution to COS variation in the boundary layer than the biogenic ones. The Paris region remains a net sink of COS even during persistent winter pollution events, because the nearest anthropogenic sources of this gas are traced to Eastern France, Germany and Benelux. In that sense it can be concluded that the recent global gridded anthropogenic emission inventory of COS [9] is only partially correct in representing those sources. In the Gif-sur-Yvette area, the nighttime net uptake of this gas remains important all year long except during late spring and early summer when a shift from sink to source is observed, the origin of which remains to be elucidated.

## Supporting information

**S1 Fig. Accuracy and long-term repeatability over 4.5 years for atmospheric COS.** The air compressed cylinder, prepared and certified by NOAA-ESRL, was analyzed by gas chromatography with PFPD detection. The GC was calibrated using a calibration gas purchased from Air Products. The LTR here expressed as the SD of the average of 96 determinations over 4.5 years is 8.6 ppt. From October 2018 to March 2019 the cylinder was not used to increase its service life.
(PDF)

**S2 Fig. Examples of 24-hour variations of COS mixing ratio and $^{222}$Rn activity in April 2015 (A) and July 2016 (B).** The night length explains why the grey band is narrower in summer than in spring. Drier conditions explain why $^{222}$Rn exhalation rates from soil are higher in summer than in spring. The blue horizontal bars indicate that $^{222}$Rn is measured in two-hour

increments and that corresponding data are reported at the beginning of the two-hour interval. Data between red brackets were averaged and the slope of linear regression during night-time inversion was calculated (n = 6 in spring, n = 5 in summer). Time is UTC.
(PDF)

**S3 Fig. Continuous measurement repeatability (CMR) of the mini-QCL.** CMR was assessed using a single target gas tank filled with dry natural air measured continuously over time periods of at least 19 h (upper panels showing the temporal variations in COS mixing ratio and laser light intensity), without (left column) and with (right column) "zero" air spectrum measurements. In the latter case, high-purity nitrogen is passed through the cell every 20 minutes for 1 minute at 500 ml/min. The lower panels present the respective Allan deviation from 1 s to $2.10^4$ s averaging time (logarithmic scale).
(PDF)

**S4 Fig. Stabilization time.** It is the time necessary to reach the final value (calculated over the last 5 min of an analysis) from CMR test with "zero" air spectrum measurements (see S3 Fig, upper right panel). The stabilization time is averaged over 25 injections from a cylinder. Also shown is the stability of the cell pressure and temperature. Cell characteristics: 76 m multi-pass cell, 500 ml of cell volume, 40 Torr of cell pressure.
(PDF)

**S5 Fig. Short-term repeatability (STR) assessment.** We evaluate the ability of the mini-QCL to always reach the same value for a target gas when alternated with a different sample, high-purity nitrogen injected for 1 min in this case. The target gas was measured 19 times and the first 4 minutes of each injection was discarded to account for stabilization. STR is the SD of 19 consecutive injections. STR = 3.6 ppt. The amplitude peak-to-peak is 13.6 ppt.
(PDF)

**S6 Fig. Long-term repeatability (LTR).** It quantifies the stability of the mini-QCL over about a month. A target gas was measured once a day alternating with ambient air measurements. The last 10 minutes of each 20 min injection were taken into account for averaging. Calibrated data are required for LTR assessment. LTR = 5.1 ppt. The amplitude peak-to-peak is 20.7 ppt.
(PDF)

**S7 Fig. Temperature dependence test.** The sensitivity of the mini-QCL to large room temperature variations (20.7–32.5 ˚C) is assessed. The top panel presents the time series of concentration (black, 1 min averages), the room temperature in the laboratory (blue) and the temperature in the cell (red). Data are expressed as deviations from the daily average. The temperature in the cell was multiplied by 10 to make the variations visible on the same scale as the room temperature (right axis). The concentration of COS is plotted against the cell temperature in the lower panel. The large COS variations (about 80 ppt peak-to-peak) cannot be corrected for changes in Tcell.
(PDF)

**S8 Fig. Water vapor sensitivity of the mini-QCL for COS data already water corrected.** All the data were averaged over 1 min. The humidifying bench is composed of one thermal mass flow controller (F-201CV, Bronkhorst), to regulate the flow of a tank filled with dry natural air, one liquid mass flow controller (Mini Cori-Flow M12, Bronkhorst), to regulate the quantity of Milli-Q water injected in the sample line, and one controlled evaporator mixer (Bronkhorst) to humidify the target gas by evaporating the water at 40 ˚C while mixing it with the gas.
(PDF)

**S9 Fig. Linearity assessment of the mini-QCL (1).** Ambient air measurement comparisons at Gif-sur-Yvette (May 2015). After water correction, the mini-QCL measurements were 5 min averaged to allow for meaningful comparisons with GC discrete measurements. The lower panel shows the linear regression between both datasets. The ordinate (4 ppt) is not significant at the level of 5% and the 95% confidence interval of the slope is 0.88–0.95.
(PDF)

**S10 Fig. Linearity assessment of the mini-QCL (2).** Performed using three air compressed cylinders, only one being Aculife-treated (NOAA-ESRL, 448.6 ± 0.2 ppt GC-MS analysis, 452.3 ± 7.0 ppt GC-PFPD analysis). The COS content of the un-treated aluminum cylinders was 526.7 ± 2.2 ppt and 729.6 ± 11.3 ppt (GC-PFPD analyses). Errors bars are 1 SD of XX min and XY replicates of compressed air measured by mini-QCL and GC, respectively.
(PDF)

**S11 Fig. Long-term repeatability (LTR) of the mini-QCL over about 7 months.** A target gas was measured once a day alternating with ambient air and calibration gas measurements (every 10 hours in the latter case). The last 10 minutes of each 20 min injection were taken into account for averaging. Data were calibrated as described in text (i.e. $[COS]_{target\ final} = 1.0547 [COS]_{dry} + B$, with $B = [COS]_{certified} - 1.0547[COS]_{raw}$ and $[COS]_{certified} = 526.7 ± 2.2$ ppt (dry gas)). The target gas was analyzed by GC-PFPD at the beginning and at the end of the survey (red dots), and in March 2017 too (744.9 ± 7.6 ppt). This demonstrates that COS can be conservative for at least 16 months also in un-treated aluminum cylinders. The failure of the mini-QCL computer is responsible for the data gap in June-July 2016.
(PDF)

**S12 Fig. Assessment of the nocturnal footprint of the GIF station.** (A) Histogram and (B) basic statistics (median, upper and lower quartiles, outliers) of nocturnal (22h–4h UTC) wind speed (measured at 10 m height) corresponding to COS exchange rates calculated using the Radon Tracer Method (n = 277).
(PDF)

**S13 Fig. Statistical distribution of net fluxes of COS at Gif-sur-Yvette.** (A) Production and uptake rates. (B) Net uptake rates of COS sorted according roughly to seasons are color coded as follows: green (spring-summer), yellow (autumn) and white (winter). 10th, 25th, 50th, median, 75th and 90th percentiles were used. Circles correspond to outliers. Until radon data for 2019 are validated, here the period of concern extends from August 2014 to December 2018.
(PDF)

**S14 Fig. Yearly variations of wind roses at the GIF site.** The period of concern is June 15 to July 15 when nocturnal Radon and COS enhancements were observed. Wind speed and direction were recorded at 10 m height.
(PDF)

**S15 Fig. Backward trajectories (3 days) ending at GIF.** Computed at 12:00 UTC, 100 m agl, using HYSPLIT's normal mode and GDAS1 meteorological data. Ending on (A) 14 February and (B) 22 March 2015.
(PDF)

**S16 Fig. Backward trajectories (3 days) ending at GIF.** Computed at 12:00 UTC, 100 m agl, using HYSPLIT's normal mode and GDAS1 meteorological data. Ending on 24 January 2016.
(PDF)

**S17 Fig. Backward trajectories (3 days) ending at GIF.** Computed at 12:00 UTC, 100 m agl, using HYSPLIT's normal mode and GDAS1 meteorological data. Ending on (A, B) 14 and 19 March 2016, respectively.
(PDF)

## Acknowledgments

SB and BL express their special thanks to Mark Zahniser at Aerodyne Research for his unconditional and enthusiastic support during operation of the mini-QCL.

## Author Contributions

**Conceptualization:** Sauveur Belviso.

**Data curation:** Sauveur Belviso.

**Funding acquisition:** Philippe Ciais.

**Investigation:** Sauveur Belviso, Benjamin Lebegue, Michel Ramonet, Victor Kazan, Isabelle Pison, Antoine Berchet, Marc Delmotte, David Montagne.

**Methodology:** Sauveur Belviso, Benjamin Lebegue, Camille Yver-Kwok.

**Software:** Camille Yver-Kwok.

**Writing – original draft:** Sauveur Belviso, Benjamin Lebegue, Michel Ramonet, Victor Kazan, Isabelle Pison, Antoine Berchet, Marc Delmotte, David Montagne.

**Writing – review & editing:** Camille Yver-Kwok, Philippe Ciais.

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
