## [Decision Letter · Decision Letter 0]

21 Nov 2019

PONE-D-19-27176

A top-down approach of sources and non-photosynthetic sinks of carbonyl sulfide from atmospheric measurements over multiple years in the Paris region (France)

PLOS ONE

Dear Dr. Belviso,

Thank you for submitting your manuscript to PLOS ONE. After careful consideration, we feel that it has merit but does not fully meet PLOS ONE’s publication criteria as it currently stands. Therefore, we invite you to submit a revised version of the manuscript that addresses the points raised during the review process.

The reviewers provided useful suggestions on improving the manuscript, especially the figures and figure captions. The method description should be amended according to the comments of Reviewer 1. We also recommend briefly discussing the issues that raised questions.

We would appreciate receiving your revised manuscript by Jan 05 2020 11:59PM. To enhance the reproducibility of your results, we recommend that if applicable you deposit your laboratory protocols in protocols.io, where a protocol can be assigned its own identifier (DOI) such that it can be cited independently in the future. For instructions see: http://journals.plos.org/plosone/s/submission-guidelines#loc-laboratory-protocols

We look forward to receiving your revised manuscript.

Kind regards,

Riikka Rinnan, Ph.D.

Academic Editor

PLOS ONE

Journal Requirements:

2. We note that Figures 1 and 7 in your submission contain [map/satellite] images which may be copyrighted. All PLOS content is published under the Creative Commons Attribution License (CC BY 4.0), which means that the manuscript, images, and Supporting Information files will be freely available online, and any third party is permitted to access, download, copy, distribute, and use these materials in any way, even commercially, with proper attribution. For these reasons, we cannot publish previously copyrighted maps or satellite images created using proprietary data, such as Google software (Google Maps, Street View, and Earth). For more information, see our copyright guidelines: http://journals.plos.org/plosone/s/licenses-and-copyright.

1. You may seek permission from the original copyright holder of Figures 1 and 7 to publish the content specifically under the CC BY 4.0 license. 

Reviewers' comments:

Reviewer's Responses to Questions

**Comments to the Author**

1. Is the manuscript technically sound, and do the data support the conclusions?

Reviewer #1: Yes

Reviewer #2: Yes

2. Has the statistical analysis been performed appropriately and rigorously? 

Reviewer #1: Yes

Reviewer #2: Yes

3. Have the authors made all data underlying the findings in their manuscript fully available?

Reviewer #1: Yes

Reviewer #2: Yes

4. Is the manuscript presented in an intelligible fashion and written in standard English?

Reviewer #1: Yes

Reviewer #2: Yes

5. Review Comments to the Author

Reviewer #1: The manuscript by Belviso et al. “A top-down approach of sources and non-photosynthetic sinks of carbonyl sulfide from atmospheric measurements over multiple years in the Paris region (France)” uses measurements of radon and atmospheric carbonyl sulfide (COS) concentrations from two different sites and few different vertical levels to determine sources and sinks of COS in the regional scale. The main finding is that due to undefined anthropogenic emissions and other undefined sources and sinks, the usage of COS measurements in determining photosynthesis is limited in the regional and global scales.

General comments:

Methods used in the study are sound but need to be clarified more in the text. Especially the “Methods” section needs to be revisited for better clarity and to help the reader. The main concern is about drawing the conclusions on emission and sink areas and processes affecting COS exchange and concentration based on the evidence showed in the manuscript. The authors should clarify the effect of boundary layer height to observations and conclusions. Also, the mechanism of how and why pollution events affect the atmospheric COS concentration remains unclear. Overall, this is an important and interesting study well suited for publication after some corrections, which are specified below.

Specific comments:

Methods: The method section should be more organized. Right now the reader gets a bit lost on when is which site or method discussed. The authors mention multiple times of calibration measurements but it is not clear whether these measurements are continuous and/or how frequent and at which site. Mention the main building/tree/vegetation height around the measurements for both sites. What is the measurement frequency of the QCL? How about averaging time? Tell more about the radon tracer method – show equation how the fluxes are determined and tell how the local exhalation rates were obtained. What is the footprint of this method and is there some quality screening and/or uncertainties related to the method? There should be a high correlation of Rn and COS concentrations for the method to work, did you check this? You could show in the supplement (correlation in addition to the time series of Fig. S1).

Results: How is the soil composition affecting the results? How about boundary layer height? How much measurement uncertainties affect the results?

Fig 1: Add the scales of the maps. Consider changing colors of the site abbreviations in the map, not clearly visible in 1B. Three shades of pink were not distinguishable on computer screen and texts not distinguishable in pdf or printed versions. From 1A you get the expression that only GIF site is there, consider making some adjustments.

Fig 2: Unclear caption. How often was the cylinder measured? Could be moved to supplementary material with other calibration/LTR/STR figures.

Fig 3: Why not use the same x-axis ticks in the different subplots? Caption for this is confusing. Check the months mentioned and also list which months are marked with white. Why are they white and not part of different seasons? Transition periods?

Fig 4: Are you sure the high positive and negative fluxes are not outliers or bad quality? Mention in the caption that these are nocturnal fluxes obtained by radon tracer method.

Fig 5: Mention which year are the measurements.

Fig 6: Proper legends needed, now the reader gets confused by lines and dots not explained in the legend. Also the caption is a bit misleading as the colors are not properly explained. The two COS signals seem to be quite similar to each other: what are the occasions when they differ and why?

Fig 7: What does the unit mean? Mention in the caption that these are during the pollution events.

Fig S1: Explain what is the grey band.

Fig S4&S5: Is this at the SAC site?

Fig S8: Use the same scales in x and y axis for clarity

Fig S9: How many analyses the three points consist of?

Fig S11: A histogram would be more representative

L22: suggest to replace “eddy flux scale” with “ecosystem scale”, also further on in the manuscript

L67-70: Mention that this applies to the regional scale

L84-87: Complicated sentence, simplify for easier read

L103: Mention which scale you are talking, seems like the sites are mostly surrounded by urban areas and croplands, while forests do not seem to be dominating

L124: How are the thresholds defined and what difference between “A” and “I”?

L129: Which GC and for which gases, is this an automated sampling system?

L147: What was calibrated? Complicated sentence.

L159: Did you follow NOAA-ESRL recommendation and use stainless steel cylinders?

L172-175: These seem more like results rather than methods

L177: “performances” refering to multiple QCL?

L199: This is repetition from L194

L205: Why this large temperature variation? In what kind of room is the QCL placed?

L205-208: Not clear whether there was a temperature dependence or not

L219: Which unit is [COS]?

L317: Do you mean that the small difference in the COS sink between seasons is unlikely observed over forests, that would have a larger difference? Currently this is not very clear.

L331-340: Where do you show evidence on solar radiation, temperature or dry conditions effect on COS exchange?

L358: From Fig. 5 it seems more that COS signal is lagging Rn signal rather than being in phase. Please check this carefully from the data. In case of a phase difference exists, how long is it?

L368: Could this be partly due to precision of measurements?

L377-380: A confusing sentence. Elsewhere?

L382-388: I believe this is mainly due to boundary layer height? In stable conditions COS is taken up by plants while emissions coming from elsewhere cannot penetrate to the shallow stable boundary layer, and thus the concentration goes down. So I am not sure you can say “anthropogenic emissions of COS are offset by natural uptake” as it is just a question of where and what height the observations are done.

L389: Write what is the emission reported by Zumkehr et al. [9]

L392-394: Not sure you can draw this conclusion based on Fig. 7. The resolution is not great and the different subplots differ from each other quite a lot.

L469: Do you have any guesses for the sources?

Reviewer #2: The Manuscript by Belviso et al. brings out quality COS data over several years in France. Apart from its importance to the sulphate budget and stratospheric aerosols, COS measurements have found renewed application in estimating the C cycle. Despite the potential of COS measurements, quality measurements around the globe characterising different sources and ecosystems are still lacking, making budget estimations difficult. In this scenario, this work is not only interesting (and sound) in its findings of different source influences but also very important from the data point of view. I recommend publication of this paper in Plos One.

Some suggestions:

Figure 1: needs to be of much better resolution, with scales, latitude, and longitude marked for the global audience. Probably modify it to show the location in a global context first (see Fig 1 of https://doi.org/10.1016/j.scitotenv.2016.02.014 for an example)

Section: COS and Rn measurements

How do you ensure that there is no contribution due to in-situ atmospheric chemistry (e.g. oxidative addition of RSCs) during or after sampling or in sample line?

Lines 226-231: it would be good to show this as a supplementary figure or by just plotting the two instruments along the same axis.

Figure 3: Looking closely, there is also a decreasing tendency in COS around early June every year. What would be a possible cause? Would it be possible to look into relationships with soil temperature, PAR etc. (you can have a look at https://doi.org/10.1016/j.scitotenv.2016.02.014 for an example)?

6. PLOS authors have the option to publish the peer review history of their article (what does this mean?). If published, this will include your full peer review and any attached files.

Reviewer #1: No

Reviewer #2: No

---

## [Author Response · Author response to Decision Letter 0]

3 Jan 2020

see attached file "response to reviewers"

---

## [Editor Report · Decision Letter 1]

15 Jan 2020

A top-down approach of sources and non-photosynthetic sinks of carbonyl sulfide from atmospheric measurements over multiple years in the Paris region (France)

PONE-D-19-27176R1

Dear Dr. Belviso,

We are pleased to inform you that your manuscript has been judged scientifically suitable for publication and will be formally accepted for publication once it complies with all outstanding technical requirements.

With kind regards,

Riikka Rinnan, Ph.D.

Academic Editor

PLOS ONE
---

## [Editor Report · Acceptance letter]

27 Jan 2020

PONE-D-19-27176R1 

A top-down approach of sources and non-photosynthetic sinks of carbonyl sulfide from atmospheric measurements over multiple years in the Paris region (France) 

Dear Dr. Belviso:

I am pleased to inform you that your manuscript has been deemed suitable for publication in PLOS ONE. Congratulations! Your manuscript is now with our production department. 

With kind regards,

on behalf of

Dr. Riikka Rinnan 

Academic Editor

PLOS ONE